# Counting Still Counts: Understanding Neural Complex Query Answering Through Query Relaxation

**Yannick Brunink***  *y.brunink@vu.nl*
*Vrije Universiteit Amsterdam*

**Daniel Daza***  *d.f.dazacruz@amsterdamumc.nl*
*Translational AI Laboratory, Department of Laboratory Medicine*
*Amsterdam University Medical Center, Vrije Universiteit Amsterdam*
*Vrije Universiteit Amsterdam*

**Yunjie He**  *yunjie.he@ipvs.uni-stuttgart.de*
*University of Stuttgart*

**Michael Cochez**  *michael.cochez@abo.fi*
*ELLIS Institute Finland & Abo Akademi University, Turku, Finland*
*Elsevier discovery lab, Amsterdam*

**Reviewed on OpenReview:** *https://openreview.net/forum?id=YVFxB6bkeC*

## Abstract

Neural methods for Complex Query Answering (CQA) over knowledge graphs (KGs) are widely believed to learn patterns that generalize beyond explicit graph structure, allowing them to infer answers that are unreachable through symbolic query processing. In this work, we critically examine this assumption through a systematic analysis comparing neural CQA models with an alternative, training-free query relaxation strategy that retrieves possible answers by relaxing query constraints and counting resulting paths. Across multiple datasets and query structures, we find several cases where neural and relaxation-based approaches perform similarly, with no neural model consistently outperforming the latter. Moreover, a similarity analysis reveals that their retrieved answers exhibit little overlap, and that combining their outputs consistently improves performance. These results call for a re-evaluation of progress in neural query answering: despite their complexity, current models fail to subsume the reasoning patterns captured by query relaxation. Our findings highlight the importance of stronger non-neural baselines and suggest that future neural approaches could benefit from incorporating principles of query relaxation.

## 1 Introduction

Knowledge graphs (KGs) encode information as entities and relations, enabling structured reasoning and efficient answering of complex queries. Traditional query answering approaches rely on symbolic processing and rule-based inference to derive answers that follow directly from the graph's explicit structure (Hogan et al., 2021). Yet, because real-world KGs are often incomplete, recent research has turned to neural methods that aim to infer likely answers beyond the observed edges (Nickel et al., 2016; Zhong et al., 2023).

In Complex Query Answering (CQA), these methods learn embeddings for entities and relations and apply neural operators to process queries in a continuous space (Ren and Leskovec, 2020; Arakelyan et al., 2021; Zhang et al., 2021; Galkin et al., 2024). The prevailing assumption is that such models can extract and combine patterns from the training graph to recover answers that are unreachable by explicit traversal

---

*Equal contribution.

alone. However, the central claim that neural reasoning truly extends the reach of symbolic reasoning has rarely been tested.

In this paper, we investigate this question directly. We compare a range of state-of-the-art neural CQA methods with a simple, training-free query relaxation strategy that broadens the constraints of a query to retrieve additional, plausible answers. This comparison allows us to test whether neural models capture information beyond what can be obtained by relaxing queries over the observed graph.

We conduct an extensive empirical study across several datasets designed to include challenging query types requiring predictions over one or more missing edges (Gregucci et al., 2025). We compare multiple state-of-the-art neural methods for CQA with a training-free query relaxation strategy. Our analysis of the results leads to three main findings: (1) **Unclear superiority.** Neural methods do not consistently outperform query relaxation strategies across datasets and query types. Although neural approaches often achieve higher performance on more complex queries, there is no single method that clearly dominates relaxation across all settings, and in several cases relaxation performs competitively or better despite involving no learning or optimization. (2) **Low answer overlap.** The top-ranked answers from neural and query relaxation approaches diverge substantially, indicating that they capture different patterns in the graph. (3) **Complementarity.** Combining their answer sets consistently improves performance, suggesting that both contribute distinct and complementary reasoning signals.

These findings challenge a fundamental assumption behind current neural CQA research: that learned representations necessarily generalize beyond symbolic reasoning. Instead, they show that simple, non-parametric query relaxation recovers a complementary set of answers that neural models often miss. This calls for a re-examination of how progress in CQA is assessed, emphasizing the need for stronger non-neural baselines and the potential of hybrid approaches that integrate symbolic relaxation mechanisms into future neural architectures.

## 2 Related Work

**Complex Query Answering.** Until recently, a large body of work on estimating missing information in KGs had been limited to the *link prediction* setting where a single relation between two entities is predicted by either learning embeddings of entities and relations, and a scoring function to compute the likelihood of a triple in a graph (Nickel et al., 2016; Wang et al., 2017; Zhong et al., 2023). Link prediction can be seen as an instance of *simple* queries, which later works extended to *complex* queries involving multi-hop prediction and variables. Methods for approximate answering of complex queries can be categorized into encoder models that encode a query into an embedding and compute an answer via similarity search (Hamilton et al., 2018a; Daza and Cochez, 2020; Chen et al., 2022; Choudhary et al., 2021; Zhang et al., 2021; Alivanistos et al., 2022; Ren et al., 2020a); and methods that traverse a computation graph following the structure of a query and produce a vector of scores for all entities in the graph (Arakelyan et al., 2021; 2023; Zhu et al., 2022; Bai et al., 2023). See (Ren et al., 2024) for a survey on different methods.

The benchmarks employed in the literature of CQA are extracted from KGs such as Freebase (Bollacker et al., 2008) and NELL (Carlson et al., 2010). They contain query-answer pairs where answers in the test set cannot be obtained by direct graph traversal. In a recent analysis of these benchmarks, Gregucci et al. (2025) conclude that they are largely dominated by *partial inference* queries, which only require predicting one missing edge in the training graph. They propose to distinguish these from *full inference* queries that require predicting all edges missing for a given query, and they demonstrate that in this setting, the performance of state-of-the-art methods for approximate query answering degrades by a large margin. Our work extends these results by further examining how the performance of these methods compares with respect to query relaxation methods that to not rely on training procedures. This provides valuable insights into how neural methods perform and how they can be improved with respect to simple heuristics.

**Statistics for predictions in KGs.** Previous works have proposed methods for learning *rules* over a KG that can be used to infer additional statements. These methods are based on enumerating subgraphs and extracting rules from them that have good coverage over the graph (Galárraga et al., 2013; Meilicke et al., 2019; Wang et al., 2024). More recently, Le et al. (2024) show that statistical features extracted from the KG

can be useful learning signals for the link prediction problem. These works provide evidence that traversing a graph and collecting statistics from it can already result in useful patterns for prediction problems on KG, which motivates our analysis of performance of neural methods for the CQA case in comparison with query relaxation strategies that can be applied to the problem.

**Query relaxation and path heuristics.** Our work is motivated by prior work on *query relaxation* in KGs. Query relaxation aims to retrieve results even when some conditions cannot be exactly met due to missing facts (Fakih and Serrano-Alvarado, 2024). Several approaches for relaxing a query exist, such as replacing constants with variables, which allow returning answers with varying degrees of exactness and ranking them by how closely they meet the original query (Hurtado et al., 2008). Other approaches are based on similarity-based relaxations, using a language model to replace entities in the query with similar ones based on the context (Mai et al., 2019); answering the query by averaging graph embeddings directly in the embedded space (Wang et al., 2018); or identifying which conditions cause empty results and incrementally broadening or dropping such constraints (Fokou et al., 2015).

Another line of related work relies on graph paths as signals for link prediction in KGs. The Path Ranking Algorithm (Lao and Cohen, 2010; Lao et al., 2011), for example, performs random walks guided by relation sequences and counts how often a node can be reached via those paths. In a similar vein, heuristics like *Katz index* (Katz, 1953) and PageRank (Page et al., 1999) which rely on existing paths in the graph can be useful for link prediction tasks (Zhang and Chen, 2018; Iyer et al., 2023).

Even though these relaxation and path-based approaches can, in principle, be applied to complex query answering, to the best of our knowledge no prior work has systematically examined whether neural methods recover the same answers, or whether they truly surpass what can be achieved by query relaxation over the graph.

## 3 Preliminaries

**KGs.** We represent a KG as a tuple $\mathcal{G} = (\mathcal{V}, \mathcal{E}, \mathcal{R})$, where $\mathcal{V}$ denotes the set of entities, $\mathcal{R}$ is the set of relations, and $\mathcal{E}$ consists of edges structured as triples $(h, r, t)$. Here, $h, t \in \mathcal{V}$ are the head and tail entities, respectively, while $r \in \mathcal{R}$ represents the relation connecting them. Each triple $(h, r, t)$ in the KG corresponds to a factual statement in first-order logic, which we express as $r(h, t)$, where $r \in \mathcal{R}$ is a binary predicate with $h$ and $t$ as its arguments.

**First-order logic queries.** A query $q$ over $\mathcal{G}$ specifies a set of constraints that an entity must fulfill to qualify as a valid answer. Entities in $\mathcal{V}$ that meet the constraints form the answer set $\mathcal{A}_q$. We consider conjunctive queries (CQs) whose constraints can be written using first order logic in disjunctive normal form. Formally, the answer set for query $q$ is the following set:

$$\mathcal{A}_q = \{v_t \mid \exists v_1, \ldots, v_N \in \mathcal{V} : (c_1^1 \wedge \cdots \wedge c_{m_1}^1) \vee \cdots \vee (c_1^n \wedge \cdots \wedge c_{m_n}^n)\}, \tag{1}$$

Here, $v_1, \ldots, v_N$ are existentially quantified variables, and $v_t$ is the *target* variable. The constraints $c_j^i$ define relationships between two variables or between a variable and a known entity (i.e. a constant) in $\mathcal{E}$, that is, $c_j^i = r(e, v)$ or $c_j^i = r(v', v)$ where $e \in \mathcal{E}$, $v, v' \in \{v_t, v_1, \ldots, v_N\}$, and $r \in \mathcal{R}$.

---

**Example 1**

Consider the query "What is the location of the baseball team for which Aaron Judge plays?". This query can be expressed as:

$$\mathcal{A}_q = \{v_t \mid \exists v_1 \in \mathcal{V} : \texttt{plays\_for}(\texttt{Aaron Judge}, v_1) \wedge \texttt{location}(v_1, v_t)\}. \tag{2}$$

This corresponds to a 2p query (query types are illustrated in Fig. 1), where the leaf node corresponds to the constant **Aaron Judge**, and the root node corresponds to the target variable $v_t$ representing a location.

---

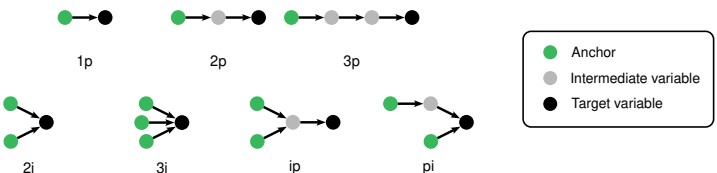

Figure 1: Query graphs for different types of conjunctive queries and their shorthand notation.

Prior work on CQA has focused on a specific subset of queries that can be associated with a *query graph*: a directed acyclic graph where leaf nodes (also known as *anchor* nodes) are constants in the query, intermediate nodes are existentially quantified variables, and the root node corresponds to the target variable. We illustrate query graphs for different types of queries in Fig. 1.

The answers to a query can be obtained by finding subgraphs of the KG that match the query graph. In the case where the graph is incomplete, several methods have been proposed for estimating *likely* answers, by training query answering models with query-answer pairs. An alternative, which to the best of our knowledge has not been explored before for the problem of CQA, is to use query relaxation strategies Fakih and Serrano-Alvarado (2024), which also allow estimating which entities are likely to be answers.

## 4 CQA via Query Relaxation

Query relaxation methods are designed to recover answers when an exact match yields few or no results due to KG incompleteness (Fakih and Serrano-Alvarado, 2024). The core idea is to iteratively relax parts of the query, such as replacing a constant entity with a variable, and then to score each entity with the counts of paths arriving to it according to the relaxed query. These relaxed matches often originate from structurally or semantically related subgraphs that would otherwise remain unreachable.

When one relaxation step is applied to a query, it is possible that multiple entities receive the same count; in this case, we say that these entities are *tied*. In our analysis, we employ a symbolic relaxation strategy that progressively relaxes a query, breaking ties in the ranking at each stage. Starting from the original query, the algorithm iteratively considers coarser relaxed forms of the query, broadening the search space while preserving its overall structure. Relaxation proceeds hierarchically: more specific matches are preferred, and further relaxation is only applied when ties occur among candidate entities. This ensures that the most constrained matches are prioritized while still providing a systematic fallback when exact matches are unavailable.

Formally, given a query $q$, the goal is to assign a score to each entity in $\mathcal{E}$ according to how well it satisfies the (possibly relaxed) query. The algorithm traverses the query graph from the leaves (constants) to the root (target variable) as follows:

1. Replace the constants at the leaves with variables, count how many paths in the KG match the relaxed query that reach each entity, and assign that count as the score.

2. When entities receive the same score (tie groups), apply a coarser relaxation to the tied portion of the query, dropping previously fixed constraints and counting matches under the simpler relaxed pattern within the tied group.

3. Residual ties are resolved by ranking entities based on their in-degree in the KG.

We present a more extensive description of the algorithm in Appendix A.1, and an example in Example 2. Among the many possible implementations of query relaxation, we adopt this strategy because it implements a fundamental capability that neural methods for query answering are *expected* to capture: the ability to exploit structural regularities and multi-hop connectivity in the graph to infer plausible answers beyond

---

**Example 2**

**Initialize**: We consider a 2p conjunctive query:

$$Aaron\ Judge \xrightarrow{PlaysBaseBallFor} ?v_1 \xrightarrow{LocatedIn} T?$$

The following triples are true but missing:

$$(Aaron\ Judge, PlaysBaseBallFor, Yankees)$$
$$(Yankees, LocatedIn, USA)$$

**Step 1: Anchor Relaxation.** Replace "Aaron Judge" with a variable and count matching paths:

$$?a_1 \xrightarrow{PlaysBaseBallFor} ?v_1 \xrightarrow{LocatedIn} T?$$

Ranking groups: (i) USA, Canada (5 paths), (ii) Mexico (3 paths), (iii) France, Spain (1 path).

$$USA = Canada \succ Mexico \succ France = Spain$$

**Step 2: Predicate-Only Relaxation (If Ties Exist).** If Step 1 still produces ties, we break them using a coarser relaxation that keeps only the final predicate:

$$?v_1 \xrightarrow{LocatedIn} T?$$

Counts (paths) : USA (100), Canada (80), France (50), Spain (50). Updated Ranking:

$$USA \succ Canada \succ Mexico \succ France = Spain$$

**Step 3: Maximum Relaxation (If Ties Persist).** Break the remaining ties using the in-degree of the tied entities:

$$?v_1 \xrightarrow{r_2} T?$$

France (500), Spain (450). Final ranking:

$$USA \succ Canada \succ Mexico \succ France \succ Spain$$

---

direct traversal. This approach thus provides a clear, interpretable baseline for testing whether learned models offer genuine advantages over training-free reasoning based solely on graph topology. Moreover, it is computationally efficient in practice, allowing a systematic comparison across large datasets and query types (we present time complexity results and runtimes in Appendix B).

## 5 Experiments

### 5.1 Methodology

**Datasets.** Earlier work has evaluated models on query answering benchmarks such as FB15k237 (Toutanova and Chen, 2015) and NELL995 (Xiong et al., 2017), introduced by Hamilton et al. (2018b), and refined by Ren and Leskovec (2020). Although these benchmarks were designed to include queries of varying structures, later analysis showed that most of their queries effectively reduce to one-hop link prediction, since answers can often be obtained by memorizing edges already present in the training graph (Gregucci et al., 2025). To overcome this limitation, we adopt the datasets introduced by Gregucci et al. (2025), which provide more balanced query sets that require predicting over one to three edges. These include FB15k237+H and NELL995+H, derived from the earlier datasets, as well as ICEWS18+H, where queries are temporally split so that test queries involve predicting future edges from past graph states.

**Evaluation metrics.** We report performance in query answering using Mean Reciprocal Rank (MRR). Given a query and a ranking of entities provided by a method, MRR is computed by taking the inverse of the ranking for a correct answer, and then averaging over all queries in the test set. We compute *filtered* MRR (Bordes et al., 2013), where for each answer the ranking is adjusted to ignore other correct answers that might be ranked higher.

**Query relaxation.** For experiments involving the query relaxation strategies described in Section 4, we load the training and validation edges in a graph database[1], which we use to answer the relaxed queries. While in principle we can compute a relaxation for every constant entity and predicate in the query, for some queries such as path queries of length three (shown as 3p in Figure 1), in practice we limit this to up to two predicates to preserve memory, which we observe to work well in practice. We refer to this method in our experimental results as RELAX. We run our experiments on a machine with an AMD EPYC-2 CPU with 128GB of RAM.

**Baselines** We run an extensive comparative analysis with several state-of-the-art neuro-symbolic CQA methods: GNN-QE (Zhu et al., 2022), ULTRAQ (Galkin et al., 2024), CQD (Arakelyan et al., 2021), ConE (Zhang et al., 2021), and QTO (Bai et al., 2023).

## 5.2 Query answering performance

Prior work on CQA has been motivated by the idea that learning-based neural methods can extrapolate beyond the information explicitly stated in the graph. Our experiments[2] are designed to test whether existing methods indeed achieve this goal, by comparing them with what can be achieved with query relaxation strategies that rely only on edges existing in the graph to score answers to a query. We thus begin our study by investigating the following research question:

> **RQ1:** Do neural methods consistently outperform a query relaxation strategy in the task of CQA over incomplete knowledge graphs?

We present results on query answering performance in Table 1, where we denote query relaxation results as RELAX. We identify several cases where RELAX performs better or comparably with neural methods. While some neural models do outperform RELAX in certain cases, this advantage is neither consistent nor predictable across datasets. For instance, ConE outperforms RELAX on the FB15k237+H dataset, but achieves similar results on 3p queries in NELL995+H, and underperforms across 1p, 2p, and 3p queries in ICEWS18+H.

We additionally observe a pattern related to the length of path queries. For 2p queries, three neural methods (GNN-QE, ULTRAQ, and ConE) perform worse than RELAX in the ICEWS18+H dataset. For 3p queries, which have an additional edge in the path, the same methods perform worse or close in the three methods, with CQD additionally performing worse in NELL995+H. These findings indicate that the gap in performance between neural methods and RELAX increases with query length, as neural methods compute predictions whose errors can propagate with length.

These findings carry particular weight given the computational overhead of neural models. They require extensive training, hyperparameter tuning, and optimization, whereas query relaxation strategies do not depend on iterative optimization and are dataset-agnostic. The narrow performance gap, and in some cases the better results obtained via relaxation, raises critical questions about the added value of current neural approaches and the computations that they might be performing. The results suggest two plausible interpretations: (i) neural models may in practice be reducing to simple statistical patterns akin to those exploited by query relaxation; or (ii) they may be retrieving fundamentally different answers.

---

[1] We use GraphDB as our graph database.
[2] All our code is available at https://github.com/yaaani85/RELAX.

Table 1: Query answering results (MRR in %). Orange indicates worse or equal performance than query relaxation strategies (RELAX), lighter orange indicates a delta of 1 or less with RELAX.

| | Model | 1p | 2p | 3p | 2i | 3i | pi | ip |
|---|---|---|---|---|---|---|---|---|
| **FB15k237+H** | GNN-QE | 42.8 | 5.2 | 4.0 | 6.0 | 8.8 | 5.6 | 9.9 |
| | ULTRAQ | 40.6 | 4.5 | 3.5 | 5.2 | 7.2 | 5.3 | 10.1 |
| | CQD | 46.7 | 4.4 | 2.4 | 11.3 | 12.8 | 6.0 | 11.5 |
| | ConE | 41.8 | 4.6 | 3.9 | 9.1 | 10.3 | 3.8 | 7.9 |
| | QTO | 46.7 | 4.9 | 3.7 | 8.7 | 10.1 | 6.1 | 13.5 |
| | RELAX | 31.5 | 4.0 | 3.9 | 5.2 | 5.2 | 1.8 | 5.3 |
| **NELL995+H** | GNN-QE | 53.6 | 8.0 | 6.0 | 10.7 | 13.3 | 16.0 | 13.5 |
| | ULTRAQ | 38.9 | 6.1 | 4.1 | 7.9 | 10.2 | 15.8 | 9.3 |
| | CQD | 60.4 | 9.6 | 4.2 | 18.5 | 19.6 | 18.9 | 22.6 |
| | ConE | 53.1 | 7.9 | 6.7 | 21.8 | 23.6 | 14.9 | 11.8 |
| | QTO | 60.3 | 9.8 | 8.0 | 14.6 | 15.8 | 17.6 | 21.1 |
| | RELAX | 35.1 | 5.6 | 5.2 | 12.4 | 12.1 | 3.9 | 6.5 |
| **ICEWS18+H** | GNN-QE | 12.2 | 0.9 | 0.5 | 16.1 | 26.5 | 19.1 | 3.5 |
| | ULTRAQ | 6.3 | 1.2 | 1.2 | 7.0 | 11.7 | 8.8 | 1.3 |
| | CQD | 16.6 | 2.5 | 1.5 | 13.0 | 19.5 | 17.1 | 6.7 |
| | ConE | 3.5 | 0.9 | 0.9 | 1.2 | 0.5 | 1.2 | 1.6 |
| | QTO | 16.6 | 2.6 | 1.4 | 15.7 | 25 | 18.4 | 6.2 |
| | RELAX | 6.1 | 1.6 | 1.3 | 1.6 | 1.0 | 1.6 | 1.6 |

> **Answer to RQ1:** Neural methods do *not* consistently outperform a query relaxation strategy in the task of CQA over incomplete knowledge graphs. In several cases, their performance is comparable or inferior. This raises the possibility that these models either collapse to count-based reasoning in complex settings, or alternatively, retrieve a different class of answers altogether.

### 5.3 Overlap Analysis

The observed comparable performance between neural models and query relaxation leads to an important question: are these methods arriving at the same answers, or do they achieve similar performance through fundamentally different predictions? Addressing this question is crucial to understanding whether neural models and symbolic approaches as the ones we investigate are truly interchangeable in this setting, or if they offer complementary capabilities. We investigate this issue in the following research question:

> **RQ2:** Is the observed similar performance between neural methods and query relaxation due to them producing the same set of answers for complex queries?

To answer this question, we examine the filtered ranks assigned to all entities by RELAX and three representative neural methods that rely on different mechanisms for query answering: ULTRAQ (Galkin et al., 2024) (which has the ability to answer queries across KGs without requiring additional training), ConE (Zhang et al., 2021) (a method that maps a query into a vector and answers queries via similarity search), and QTO (Bai et al., 2023) (a neuro-symbolic method that traverses the query graph with fuzzy operators). Given a query and a pair of rankings lists produced by RELAX and a neural model, we collect the set of entities in the top $k$ for each method. Let $A_R^{(k)}$ and $A_N^{(k)}$ be such sets for RELAX and a neural model, respectively, for a given value of $k$. To measure the overlap among these sets, we compute the Jaccard similarity at $k$, which we define as

$$\text{J@}k = \frac{|A_R^{(k)} \cap A_N^{(k)}|}{|A_R^{(k)} \cup A_N^{(k)}|}. \tag{3}$$

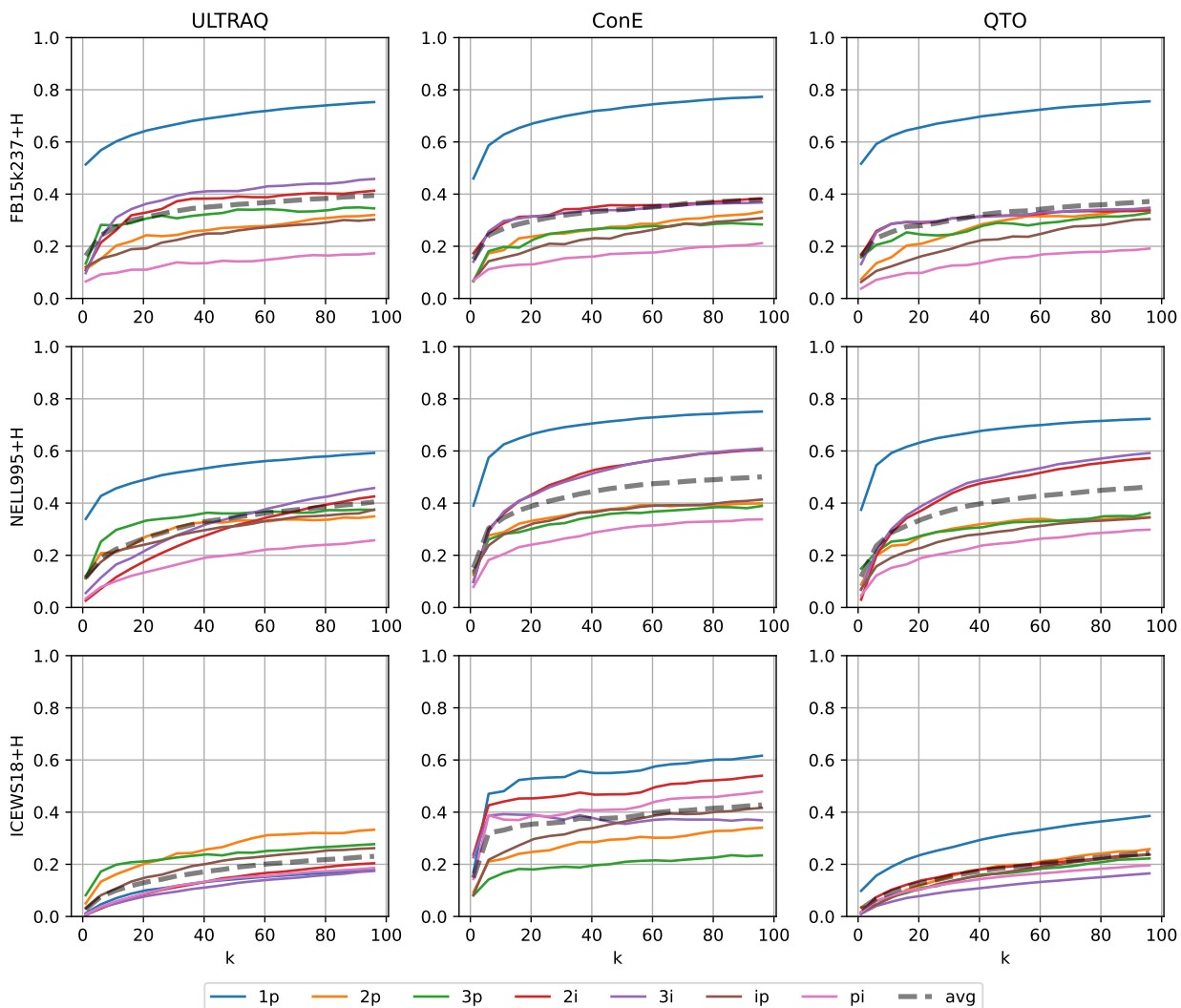

Figure 2: Jaccard similarity of top-$k$ retrieved answers between RELAX and three neural methods (columns), across different datasets (rows) and query structures (shown in colors in the legend). If the top-k contains the same set of answers, the similarity would reach 1.

According to this definition, a value of J@$k$ of 1 indicates that the two methods rank the same entities in the top $k$, whereas a value of 0 means that the sets are disjoint.

We present results of J@$k$ for different values of $k$ in Fig. 2. In the FB15k237+H and NELL995+H datasets, for simple query types such as 1p, RELAX and neural models exhibit moderate to high overlap, approaching from 60% to up to 80% for high values of $k$. However, as query structure complexity increases (e.g., 2i, 3p, pi), overlap declines substantially. For ICEWS18+H, the overlap is much lower (with ULTRAQ showing the lowest degree of similarity). These results indicate that while neural models and RELAX may achieve similar MRR, they do so by ranking different entities highly. This divergence is especially notable given the simplicity of RELAX, which relies on symbolic path enumeration and counting. That a neural model, despite being far more complex, retrieves a distinct set of answers suggests that it fails to capture certain signals that RELAX exploits, and vice versa.

> **Answer to RQ2:** The similar performance between neural models and query relaxation does not stem from retrieving the same answers. Their top-ranked outputs diverge significantly for complex queries, suggesting that these methods capture complementary patterns.

### 5.4 Combining neural and symbolic methods

The divergence in top-ranked answers between neural models and query relaxation, particularly for complex query structures, suggests that these methods might be capturing complementary patterns that are useful for the task. We therefore investigate the following research question:

> **RQ3:** Is there an optimal combination of answers computed by neural methods and query relaxation that consistently produces better results than their individual performance in the task of complex query answering?

To answer this question, we introduce an oracle-based evaluation aimed at determining the theoretical optimal combination of RELAX and several neural methods. For each query, the oracle selects the list of scores from each method that yields the higher MRR for the given query.

We present the results for QTO, query relaxation, and their oracle combination (denoted RELAX+QTO) in Table 2. Statistical significance is assessed using paired $t$-tests with Holm–Bonferroni correction for multiple comparisons, and all differences between RELAX+QTO and the individual methods are significant ($p < 0.001$). Across datasets and query types, the oracle combination consistently achieves higher MRR, with relative gains in the metric reaching **30–50%** for several query types (notably 2p–3p in FB15k237+H and NELL995+H).

A closer inspection reveals clear patterns in where these gains arise. On average over the three datasets, the largest improvements occur for **path queries** (2p, 3p), which require reasoning over longer relational chains. This suggests that query relaxation effectively recovers structural connections that neural models only partially capture or whose learned representations degrade with path length. In contrast, for **intersection queries** (denoted as 2i and 3i in Figure 1), the relative improvements are smaller, particularly in ICEWS18+H. This dataset is temporally split and generally more challenging, and both neural and RELAX appear less effective.

When extending the analysis to other neural methods (see Appendix C), we observe similar trends. ConE benefits across all query types but especially on path queries, reinforcing the idea that relaxation complements path-based reasoning. ULTRAQ, a foundation model capable of zero-shot transfer across KGs, shows the strongest relative gains overall, indicating that large, pretrained neural architectures may still overlook local structural regularities that relaxation can recover efficiently.

Overall, these results are evidence of systematic complementarity between neural reasoning and query relaxation. The distinct patterns across query types, datasets, and architectures highlight where current neural models fall short, particularly on path queries and temporally complex graphs, and point to opportunities for integrating relaxation-like mechanisms into future neural CQA approaches.

> **Answer to RQ3:** The optimal combination of neural and query relaxation predictions yields consistently and significantly gains across datasets and query types (particularly for path queries), demonstrating that both approaches capture complementary reasoning signals, rather than redundant ones.

## 6 Discussion

Our analyses aim to understand what current neural methods for complex query answering (CQA) over knowledge graphs actually capture. By comparing them with a training-free query relaxation strategy that exploits the graph structure alone, we evaluate whether neural models can compute answers beyond those

Table 2: Query answering results (MRR) comparing the original performance of QTO, RELAX, their optimal combination, denoted as RELAX+QTO, and the relative difference in percent.

| Model | 1p | 2p | 3p | 2i | 3i | pi | ip |
|---|---|---|---|---|---|---|---|
| **FB15k237+H** | | | | | | | |
| QTO | 46.7 | 5.3 | 3.8 | 8.9 | 10.6 | 6.1 | 13.9 |
| RELAX | 31.5 | 4.0 | 3.9 | 5.2 | 5.2 | 1.8 | 5.3 |
| RELAX+QTO | **49.0** | **7.1** | **5.6** | **10.9** | **12.2** | **6.8** | **15.7** |
| Abs. $\Delta$ | 2.4 | 1.9 | 1.7 | 2.0 | 1.6 | 0.7 | 1.8 |
| Rel. $\Delta$ (%) | 5.1 | 35.1 | 44.6 | 22.3 | 15.6 | 11.4 | 12.8 |
| **NELL995+H** | | | | | | | |
| QTO | 60.7 | 10.0 | 7.6 | 15.6 | 17.0 | 17.0 | 21.3 |
| RELAX | 35.1 | 5.6 | 5.2 | 12.4 | 12.1 | 3.9 | 6.5 |
| RELAX+QTO | **64.9** | **12.7** | **9.9** | **23.6** | **23.7** | **18.1** | **23.5** |
| Abs. $\Delta$ | 4.2 | 2.7 | 2.3 | 8.0 | 6.7 | 1.1 | 2.2 |
| Rel. $\Delta$ (%) | 7.0 | 27.1 | 30.8 | 51.2 | 39.1 | 6.4 | 10.3 |
| **ICEWS18+H** | | | | | | | |
| QTO | 16.6 | 2.6 | 1.3 | 14.8 | 22.5 | 17.2 | 5.9 |
| RELAX | 6.1 | 1.6 | 1.3 | 1.6 | 1.0 | 1.6 | 1.6 |
| RELAX+QTO | **19.3** | **3.5** | **2.2** | **15.4** | **22.7** | **17.8** | **6.8** |
| Abs. $\Delta$ | 2.7 | 1.0 | 0.9 | 0.6 | 0.2 | 0.6 | 0.9 |
| Rel. $\Delta$ (%) | 16.2 | 38.2 | 67.5 | 3.8 | 1.0 | 3.4 | 15.7 |

reachable by symbolic reasoning. Across datasets and query types, neural methods do not consistently outperform query relaxation. In many cases, their performance is similar or lower, which does not rule out that learned representations may often reproduce patterns that can already be recovered through structural statistics.

However, a more detailed analysis shows that the low overlap between neural and query relaxation methods, coupled with the strong improvements observed when combining them, indicates that both approaches capture complementary reasoning signals. Neural models and query relaxation thus emphasize different aspects of the graph which together explain a broader range of answers. These findings motivate rethinking the relationship between symbolic and neural reasoning in CQA as interacting perspectives that can inform future hybrid approaches.

Concretely, we find that a promising area where neural methods and query relaxation complement each other is in path queries. We observe that the relative improvements of the optimal combination of the two approaches increases consistently with query length. This suggests that the compounding errors, which we observe in prior work (Hamilton et al. (2018a); Ren et al. (2020b); Ren and Leskovec (2020), among others) can be improved by incorporating information available from query relaxation methods. This interpretation is further supported by an analysis of successive relaxation steps for 2p queries (Appendix C), which shows that performance is preserved or improved at each step, highlighting that query relaxation mitigates error accumulation by only refining scores within tie groups.

## 6.1 Limitations and Future Work

Our work relies on query relaxation as an analytical tool to understand the comparative performance of neural methods for CQA, which is effective for the scale of KGs that we consider in this work and that have been used to benchmark existing neural methods. Extending this analysis to large scale KGs might require additional techniques of scalability, and especially if we were to study more complex queries than

the ones considered in this work. An example, which we implemented in our work, is limiting relaxations to a subset of predicates. For larger scale KGs, future analyses should explore principled ways of pruning or approximating relaxations while retaining predictive power, especially when considering incorporating the learnings from our work into future models for CQA.

Our combination experiments used an oracle to estimate the theoretical upper bound of complementarity. This is useful for analysis purposes, but designing realistic fusion mechanisms, such as learned ensembles, meta-predictors, or neural architectures that explicitly incorporate relaxation signals, remains an open avenue. Concretely, we note two concrete directions for designing future methods based on our results: (1) **Heuristic routing**, where query relaxation counts are used when their signal is strong (for example, few ties remaining after a relaxation step), otherwise falling back to a neural model when query relaxation produces large tie groups. In this sense, query relaxation provides a notion of confidence based on how discriminative counts are for a given query. The challenge of implementing such an approach is determining precise strategies for choosing one approach over the other. (2) **Rank-level ensembling**, where normalized ranks or scores from query relaxation and neural models are combined using learned weights. This approach requires devising principled combination methods, as RELAX computes counts, which have a different interpretation than the normalized scores computed by neural models.

Such designs would require identifying which aspects of query relaxation methods can be usefully integrated into a learning pipeline, and how to do so without sacrificing differentiability or scalability.

Finally, our analysis focuses on one instantiation of a broader class of query relaxation strategies. Future work could explore alternative relaxation schemes, richer statistical features (e.g., path diversity, edge weights, type constraints), or domain-informed heuristics. Such extensions may further sharpen our understanding of what current neural methods capture and what they miss.

## 7 Conclusion

Our study provides a systematic analysis of neural methods for complex query answering through the lens of query relaxation. Across diverse datasets and query types, we find that neural approaches do not always retrieve the same answers than a simple, training-free relaxation strategy. Despite comparable overall performance, their retrieved answers differ substantially, and combining both yields statistically significant improvements. These findings highlight the importance of strong, interpretable non-neural baselines in assessing progress in CQA, and reveal that neural and symbolic reasoning capture complementary patterns in the graph. Understanding and bridging this gap may guide the next generation of neural architectures toward models that integrate the structural reasoning capabilities of symbolic approaches with the flexibility of learned representations.

## Acknowledgments

For this work, Michael Cochez is partially funded by the Elsevier Discovery Lab, partially funded by the Graph-Massivizer project, funded by the Horizon Europe programme of the European Union (grant 101093202), and supported by a gift from Accenture LLP. His work on this publication is in part based upon work from COST Action CA23147 GOBLIN - Global Network on Large-Scale, Cross-domain and Multilingual Open Knowledge Graphs, and COST Action CA24121 - Knowledge Graphs in the Era of Large Language Models (KGELL), both supported by COST (European Cooperation in Science and Technology, https://www.cost.eu). Daniel Daza was funded by a gift from Accenture LLP.

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

## A  Appendix

### A.1  Query relaxation strategies

In Algorithm 1 we present the exact procedure used to obtain the results reported in the main paper. Next, we motivate our design decisions and discuss worst case complexity and run time.

### A.2  Design Decisions

**Progressive Relaxation**  Our relaxation strategy follows a progressive relaxation philosophy, prioritizing signals that preserve the most specific query information before resorting to more general cues. While fully relaxed features (e.g., global indegree) can indeed be strong indicators of prominence or connectivity, they are also less discriminative with respect to the original query intent. By starting from the most constrained (anchor-relaxation-only) formulation and introducing increasingly relaxed criteria only as tie-breakers, we ensure that ranking decisions remain grounded in the semantics of the original query. The database literature offers many alternative ways to combine relaxed signals (Fakih and Serrano-Alvarado, 2024; Hurtado et al., 2008; Mai et al., 2019; Fokou et al., 2015), but our approach is guided by the philosophy that more specific constraints should always dominate less specific ones, as they carry higher semantic fidelity to the original query.

In our experiments, this progressive strategy consistently yielded the most stable and interpretable results, even though we do not claim it to be universally optimal. Instead, our goal is to compare such strategies with existing neural methods for query answering.

**Fallback Strategy**  The final two stages of progressive relaxation rely on global signals, specifically, (relation) indegrees, that do not depend on the structure of the query. We therefore precompute and store these values in a lookup table, enabling fast retrieval at query time. This design allows us to produce rankings even when earlier, more specific query evaluations are skipped due to space or time constraints, ensuring both efficiency and robustness.

**Efficiency Tradeoff**  We observe that, given our anchor-relaxation query and available statistics, only the $3p$ query has an additional relaxation step that could be used as an additional tiebreaker by re-executing this further relaxed query to the engine. We choose not to do this in favor of a clear and efficient workflow: the query engine is invoked only once (the expensive step), and all subsequent relaxations rely on lightweight statistical lookups.

**Query Decomposition Trick**  For intersection-type queries (e.g., 2i, 3i), direct evaluation can produce a large intermediate join on the shared variable (e.g., target $t$), yielding cubic or quartic complexity. To mitigate this, we evaluate each arm of the intersection separately as an `aggregate` subquery grouped by $t$, and then merge the resulting tables. We use this optimized version for intersection-only queries but keep the original (non-aggregated) formulation for projection–intersection combinations.

## B  Complexity Analysis

Relaxation proceeds hierarchically: more specific matches are preferred, and further relaxation is only applied when ties occur among candidate entities worst-case complexity of Step 2 (anchor-relaxation) of Algorithm 1. We evaluate each query type on a Knowledge Graph (KG) with $N_e$ entities and $N_r$ relations. Complexities are expressed in terms of $N_e$ since for KGs it dominates the cost of enumerating possible bindings. Each query pattern can be seen as a join of triple patterns; the number of free (unbound) entity variables determines the exponent of $N_e$ in the worst case. Dense graphs, where each entity is connected to every other entity by each relation type, represent the upper bound for these operations. Since the intermediate join results must be materialized (or at least streamed) to compute counts, the space complexity scales in the same order as time.

Therefore, for a $k$-hop path or multi-intersection query with worst-case complexity $O(N_e^k)$, both the time and space complexities are (generally) $O(N_e^k)$.

For intersection queries, we mitigate this by decomposing the complex query in multiple simple queries and aggregate the results. See section A.2. This *subquery aggregation trick* reduces the overall complexity (e.g., from $O(N_e^3)$ to $O(3N_e)$). For all other query types, the asymptotic time complexity does corresponds to the number of possible entity bindings explored during query execution, see Table 3.

However, in practice KGs are highly sparse graphs, and the worst-case complexity is not a realistic scenario. Therefore, we also provide an empirical evaluation of the runtimes, which we present next.

### B.1 Empirical Run Time Performance

We evaluate run time performance on the benchmarks used in this study. We set the standard GraphDB maximum memory to 10GB (default), and we enforce a timeout of 10s. Naturally, both can adjusted based on preferences. Please present results in Table 4. The results are in line with the theoretical complexity analysis (Table 3) and confirm the practical observation that KGs are sparse graphs i.e., our fallback strategy only has to be invoked an acceptable number of timeouts. Our primary goal was the analysis of the results, not runtime performance. If that is a concern, one could execute queries in parallel to improve efficiency.

## C Additional Results

**Alternative metrics.** As an alternative to MRR, we present results of Hits at 10 (H@10) in Table 5. Similar to the results observed with MRR, we notice that there in several cases, the performance of RELAX is close or better than that of neural methods.

**Combining RELAX with other methods.** Section 5.4 shows the upper bound in performance that results from the optimal combination of rankings produced by QTO and RELAX. Here we present additional results where RELAX is combined with ConE (Table 6) and ULTRAQ (Table 7). Similar to the experiments with QTO, we note that the upper bound results in a significant increase in MRR, which demonstrates the complementary nature of the correct answers computed by neural methods and RELAX.

**Effect of progressive query relaxation.** To further analyze the behavior of query relaxation and assess whether progressively relaxing a query leads to error escalation, we examine the effect of individual relaxation steps on performance. In particular, we focus on path queries of length two (denoted as 2p), for which the relaxation procedure proceeds through multiple stages as described in Section 4.

Recall that the relaxation strategy operates hierarchically: at each step, only entities that received identical scores in the previous step (tie groups) are affected, while the relative ordering of entities with distinct scores is preserved. This design suggests that additional relaxation steps should not degrade performance, but instead either maintain or improve the ranking of correct answers. To verify this empirically, we compute Mean Reciprocal Rank (MRR) after each relaxation step for 2p queries on FB15k237+H, NELL995+H, and ICEWS18+H. The results are shown in Table 8.

Across all datasets, we observe that intermediate relaxation steps preserve MRR, indicating that breaking ties at these stages has no effect on performance. Importantly, the final relaxation step consistently increases MRR, demonstrating that deeper relaxation resolves ambiguity among tied candidates rather than introducing additional error. These results provide no evidence of error escalation as relaxation proceeds, and instead support the stability of the hierarchical relaxation strategy. They further contrast with neural query answering methods, where errors may accumulate as predictions are composed over longer paths.

**Preliminaries / Definitions:**

- $\mathcal{G} = (\mathcal{E}, \mathcal{R})$: Knowledge graph with edges $\mathcal{E}$ and relations $\mathcal{R}$

- $Q$: Conjunctive query over $\mathcal{G}$

- Anchors $\mathcal{A} = \{a_1, \ldots, a_k\}$: known entities in $Q$

- Target variable $v_t$: variable in $Q$ to rank candidate answers for

- RelaxAnchors$(Q)$: Replace anchors $a_i \in Q$ with newly coined variables $?a_i$

- $\mathsf{InDeg}_r(x) := \left| \{\, y \in \mathcal{V} \mid (y, r, x) \in \mathcal{E} \,\} \right|$ (indegree w.r.t relation)

- $\mathsf{InDeg}(x) := \left| \{\, (y, r) \in \mathcal{V} \times \mathcal{R} \mid (y, r, x) \in \mathcal{E} \,\} \right|$ (total indegree)

- $v_t$ satisfies $Q \iff \exists$ path(s) in $\mathcal{G}$ matching the constraints of $Q$

- For intersection queries of the form, we define
  $$\pi_i(v_t) \leftarrow |\{(a_i, r_i, v_t) \in \mathcal{E}\}_{i=1}^k|$$

**Input:** Conjunctive Query $Q$ with anchors $\mathcal{A} = \{a_1, \ldots, a_k\}$, a KG $\mathcal{G}$, target variable $v_t$
**Output:** Ranked candidate answers for $v_t$

**Step 1: Anchor relaxation**

- $Q_{\text{relaxed}} \leftarrow \mathsf{RelaxAnchors}(Q)$

**Step 2: Execute relaxed query and initial scoring**

- $C \leftarrow \{v_t \mid v_t \text{ satisfies } Q_{\text{relaxed}} \text{ in } \mathcal{G}\}$

- If $Q$ is intersection query:

  - For each anchor $a_i$, compute $\pi_i(v_t)$
  - $\text{score}(v_t) \leftarrow \prod_{i=1}^k \pi_i(v_t)$

- Else:

  - $\text{score}(v_t) \leftarrow$ number of matching paths

- Partition candidates into tie groups:

  $$G_1 \succ G_2 \succ \cdots \succ G_m, \quad G_j = \{c \in C \mid \text{same Step 2 score}\}$$

**Step 3: In-degree given relation (for ties within groups)**

- For each tie group $G_j$:

  - If $Q$ is intersection: $\text{score}(c) \leftarrow \prod_{i=1}^k \mathsf{InDeg}_{r_i}(c), \forall c \in G_j$
  - Else: $\text{score}(c) \leftarrow \mathsf{InDeg}_r(c), \forall c \in G_j$

- Higher-ranked groups remain above lower-ranked groups

**Step 4: Target indegree (maximum relaxation for remaining ties)**

- For each tie group still tied: $\text{score}(c) \leftarrow \mathsf{InDeg}(c), \forall c \in G_j$

**Return:** Candidates $v_t$ sorted by descending score$(v_t)$, preserving group order

**Algorithm 1:** RELAX: Hierarchical Query Relaxation for Conjunctive Queries with Grouped Tie-Breaking

Table 3: Worst-case time complexity for each query type when the anchor is relaxed. Complexity grows with the number of free entity variables. For intersection queries, subquery aggregation significantly reduces cost.

| Query type | Relaxed pattern | Free vars | Evaluation style | Worst-case |
|---|---|---|---|---|
| 1p | $(?a, r, T)$ | 1 | table lookup | $O(c)$ |
| 2p | $(?a, r_1, x) \wedge (x, r_2, T)$ | 2 | direct | $O(N_e^2)$ |
| 3p | $(?a, r_1, x_1) \wedge (x_1, r_2, x_2) \wedge (x_2, r_3, T)$ | 3 | direct | $O(N_e^3)$ |
| 2i | $(?a_1, r_1, T) \wedge (?a_2, r_2, T)$ | 2 | with aggregation trick | $O(2N_e)$ |
| 3i | $(?a_1, r_1, T) \wedge (?a_2, r_2, T) \wedge (?a_3, r_3, T)$ | 3 | with aggregation trick | $O(3N_e)$ |
| ip | $(?a_1, r_1, t) \wedge (?a_2, r_2, t) \wedge (t, r_3, T)$ | 3 | direct | $O(N_e^3)$ |
| pi | $(?h, r_1, x) \wedge (x, r_2, T) \wedge (?a, r_3, T)$ | 3 | direct | $O(N_e^3)$ |

Table 4: Empirical run time performance for each query type when the anchor is relaxed (avg, std, timeout ratio). A timeout is called if the query does not finish within 10 seconds.

| Dataset | 1p | 2p | 3p | 2i | 3i | ip | pi |
|---|---|---|---|---|---|---|---|
| **FB15k237+H** | | | | | | | |
| avg (ms) | 30 | 80 | 331 | 60 | 72 | 1187 | 276 |
| std (ms) | 54 | 129 | 559 | 56 | 65 | 1963 | 579 |
| timeout (%) | 0 | 0 | 0 | 0 | 0 | 15.6 | 0 |
| **NELL995+H** | | | | | | | |
| avg (ms) | 33 | 47 | 107 | 50 | 56 | 1088 | 95 |
| std (ms) | 112 | 37 | 268 | 61 | 74 | 1969 | 158 |
| timeout (%) | 0 | 0 | 0 | 0 | 0 | 7.8 | 0 |
| **ICEWS18+H** | | | | | | | |
| avg (ms) | 127 | 321 | 1382 | 146 | 143 | 84 | 1182 |
| std (ms) | 288 | 290 | 1786 | 142 | 135 | 625 | 1757 |
| timeout (%) | 0 | 0 | 1.9 | 0 | 0 | 0.5 | 2.0 |

Table 5: Query answering results (H@10 in %). Orange indicates worse or equal performance than RELAX, lighter orange indicates a delta of 1 or less with RELAX.

| | Model | 1p | 2p | 3p | 2i | 3i | ip | pi |
|---|---|---|---|---|---|---|---|---|
| **FB15k237+H** | ULTRAQ | 59.8 | 9.5 | 6.7 | 10.2 | 12.9 | 15.7 | 8.4 |
| | ConE | 58.4 | 7.0 | 5.8 | 16.2 | 19.6 | 10.1 | 6.3 |
| | QTO | 67.1 | 9.7 | 6.7 | 18.0 | 19.8 | 21.9 | 10.7 |
| | RELAX | 45.9 | 7.3 | 6.5 | 8.8 | 9.7 | 8.5 | 3.1 |
| **NELL995+H** | ULTRAQ | 50.7 | 11.9 | 9.0 | 14.4 | 19.2 | 17.7 | 26.3 |
| | ConE | 71.8 | 15.3 | 10.5 | 40.7 | 45.1 | 21.4 | 24.6 |
| | QTO | 77.3 | 18.1 | 12.5 | 32.3 | 34.4 | 31.3 | 29.7 |
| | RELAX | 51.8 | 10.0 | 9.7 | 25.3 | 26.1 | 11.5 | 8.1 |
| **ICEWS18+H** | ULTRAQ | 12.2 | 2.3 | 2.4 | 13.4 | 20.5 | 2.8 | 17.0 |
| | ConE | 8.7 | 1.9 | 1.3 | 2.3 | 1.0 | 3.1 | 2.4 |
| | QTO | 32.5 | 4.9 | 2.4 | 27.1 | 33.7 | 11.7 | 29.9 |
| | RELAX | 11.8 | 2.9 | 2.2 | 3.5 | 2.0 | 3.2 | 3.3 |

Table 6: Query answering results (MRR) comparing the original performance of ConE, RELAX, their optimal combination, denoted as RELAX+ConE, and the relative difference in percent.

| Model | 1p | 2p | 3p | 2i | 3i | pi | ip |
|---|---|---|---|---|---|---|---|
| **FB15k237+H** | | | | | | | |
| ConE | 38.1 | 3.6 | 3.0 | 7.8 | 9.7 | 3.2 | 5.5 |
| RELAX | 31.5 | 4.0 | 3.9 | 5.2 | 5.2 | 1.8 | 5.3 |
| RELAX+ConE | **44.2** | **5.7** | **5.4** | **10.0** | **11.5** | **4.0** | **8.2** |
| Abs. Δ | 6.1 | 1.8 | 1.5 | 2.2 | 1.8 | 0.8 | 2.7 |
| Rel. Δ (%) | 16.0 | 44.5 | 38.0 | 27.6 | 18.5 | 24.7 | 49.0 |
| **NELL995+H** | | | | | | | |
| ConE | 52.9 | 7.8 | 5.7 | 19.5 | 22.5 | 14.0 | 11.9 |
| RELAX | 35.1 | 5.6 | 5.2 | 12.4 | 12.1 | 3.9 | 6.5 |
| RELAX+ConE | **58.4** | **10.1** | **8.4** | **25.1** | **27.1** | **15.1** | **14.2** |
| Abs. Δ | 5.6 | 2.3 | 2.7 | 5.6 | 4.6 | 1.1 | 2.3 |
| Rel. Δ (%) | 10.5 | 29.2 | 46.8 | 28.5 | 20.5 | 8.2 | 19.0 |
| **ICEWS18+H** | | | | | | | |
| ConE | 3.8 | 0.9 | 0.8 | 1.0 | 0.4 | 1.1 | 1.4 |
| RELAX | 6.1 | 1.6 | 1.3 | 1.6 | 1.0 | 1.6 | 1.6 |
| RELAX+ConE | **7.1** | **2.0** | **1.7** | **1.8** | **1.1** | **1.9** | **2.3** |
| Abs. Δ | 0.9 | 0.4 | 0.4 | 0.2 | 0.1 | 0.3 | 0.7 |
| Rel. Δ (%) | 14.8 | 24.8 | 26.9 | 11.8 | 9.2 | 20.3 | 44.0 |

Table 7: Query answering results (MRR) comparing the original performance of ULTRAQ, RELAX, their optimal combination, denoted as RELAX+ULTRAQ, and the relative difference in percent.

| Model | 1p | 2p | 3p | 2i | 3i | pi | ip |
|---|---|---|---|---|---|---|---|
| **FB15k237+H** | | | | | | | |
| ULTRAQ | 40.6 | 4.5 | 3.5 | 5.2 | 7.2 | 5.3 | 10.1 |
| RELAX | 31.5 | 4.0 | 3.9 | 5.2 | 5.2 | 1.8 | 5.3 |
| RELAX+ULTRAQ | **45.6** | **6.3** | **5.3** | **8.3** | **9.8** | **6.1** | **11.6** |
| Abs. Δ | 4.9 | 1.8 | 1.4 | 3.1 | 2.6 | 0.8 | 1.5 |
| Rel. Δ (%) | 12.1 | 41.3 | 36.6 | 59.5 | 35.9 | 14.8 | 15.1 |
| **NELL995+H** | | | | | | | |
| ULTRAQ | 38.9 | 6.1 | 4.1 | 7.9 | 10.2 | 15.8 | 9.3 |
| RELAX | 35.1 | 5.6 | 5.2 | 12.4 | 12.1 | 3.9 | 6.5 |
| RELAX+ULTRAQ | **52.4** | **8.8** | **7.0** | **18.3** | **19.2** | **17.3** | **12.2** |
| Abs. Δ | 13.5 | 2.6 | 1.8 | 5.9 | 7.0 | 1.5 | 2.9 |
| Rel. Δ (%) | 34.8 | 43.0 | 34.1 | 47.1 | 58.2 | 9.8 | 31.4 |
| **ICEWS18+H** | | | | | | | |
| ULTRAQ | 6.3 | 1.2 | 1.2 | 7.0 | 11.7 | 8.8 | 1.3 |
| RELAX | 6.1 | 1.6 | 1.3 | 1.6 | 1.0 | 1.6 | 1.6 |
| RELAX+ULTRAQ | **11.6** | **2.3** | **2.0** | **8.1** | **12.1** | **9.8** | **2.4** |
| Abs. Δ | 5.3 | 0.7 | 0.6 | 1.1 | 0.5 | 1.0 | 0.8 |
| Rel. Δ (%) | 85.2 | 44.0 | 47.7 | 15.3 | 3.9 | 11.7 | 48.6 |

Table 8: MRR after successive relaxation steps for 2p queries.

| Relaxation step | FB15k237+H | NELL995+H | ICEWS18+H |
|---|---|---|---|
| 1 | 0.0315 | 0.0428 | 0.0148 |
| 2 | 0.0315 | 0.0428 | 0.0148 |
| 3 | 0.0401 | 0.0565 | 0.0160 |

