# OpenReview forum: "Counting Still Counts: Understanding Neural Complex Query Answering Through Query Relaxation"
_TMLR — Accepted by TMLR_

### Review · Reviewer_FBxT · 2025-12-18

**Summary Of Contributions:**

**Summary of Contributions:** This work challenges a prevailing assumption in neural CQA research and highlights the importance of strong, interpretable non-neural baselines. It also suggests that future progress may benefit from hybrid approaches that integrate structural query relaxation principles into neural architectures.

**Strengths:**
- Rather than proposing yet another neural model, the paper asks a well-defined and underexplored question about what existing neural CQA methods actually capture, making the contribution conceptually valuable.
- The findings provide concrete evidence for complementarity between neural and symbolic reasoning, offering useful guidance for future hybrid methods.

**Weaknesses:**
- The oracle fusion does not translate directly into a practical method, and the paper does not propose a realizable hybrid approach
- How broadly the conclusions extend to more expressive query classes or alternative relaxation schemes? Now, the analysis focuses on conjunctive queries and a specific instantiation of query relaxation

**Audience:**

Yes

**Audience Explanation:**

While the work is specialized to CQA over knowledge graphs, its methodological insights, especially regarding how to rigorously assess claims of generalization and reasoning, are likely to resonate with readers interested in trustworthy evaluation and hybrid reasoning systems

**Broader Impact Concerns:**

The submission focuses on a methodological and empirical analysis of complex query answering over knowledge graphs. It does not introduce new data sources, deployable systems, or application-specific models that could directly raise ethical concerns related to fairness, privacy, security, or misuse. The proposed query relaxation strategy is analytical in nature and is used primarily as a diagnostic baseline rather than as a production-ready system.

While the findings may influence future research directions in neural–symbolic reasoning and benchmarking practices, these implications are methodological rather than societal. As such, the work does not appear to require an additional Broader Impact Statement beyond standard disclosure, and there are no ethical risks that are insufficiently addressed.

**Claims And Evidence:**

Yes

**Claims Explanation:**

The paper’s central claims that neural CQA methods do not consistently outperform a training-free query relaxation baseline, that the two approaches retrieve substantially different answers, and that their combination yields complementary gains are each backed by targeted empirical analyses. The authors evaluate multiple SOTA neural models across several datasets and query structures, using standard metrics, which lends credibility to the performance comparisons.

Beyond aggregate performance, the overlap analysis based on Jaccard similarity provides convincing evidence that similar MRR scores often arise from fundamentally different ranked answers, directly supporting the claim that neural methods do not subsume relaxation-based reasoning. The oracle-based combination experiments further strengthen this conclusion by demonstrating consistent and statistically significant gains, reinforcing the argument for complementarity rather than redundancy.

While the analysis is limited to conjunctive queries and a specific relaxation strategy, within this scope the experimental design is careful, the evidence is coherent, and the conclusions are well-aligned with the presented results.

**Requested Changes:**

- I recommend the authors try to discuss practical hybrid strategies beyond the oracle. The oracle-based combination experiments convincingly demonstrate complementarity,  while plausible non-oracle fusion strategies, e.g., heuristic selection rules, confidence-based routing, or lightweight learned combinations, even if left for future work, would make the implications of the results more concrete.
- Please provide additional intuition for failure modes of neural methods.  The analysis shows that neural performance degrades with query length and that overlap with relaxation decreases for more complex queries. A more explicit discussion, possibly with qualitative examples, of why certain neural models fail in these settings would further enhance interpretability.
- Add brief reminders of query-type notation (e.g., 2p, 3i) when first referenced in the experimental section.
- Ensure consistent terminology when referring to “query relaxation,” “relaxation-based reasoning,” and “RELAX” to avoid any possible confusion for non-specialist readers.

---

> ### Author Response · Authors · 2026-02-06
>
> We thank the reviewer for their careful reading, the positive assessment of our contributions, and their constructive suggestions. In our updated version, we have incorporated the requested changes (RC), which we detail next.
>
> **RC1:** While the oracle experiment is intended to estimate an upper bound on complementarity, we agree that more concrete directions can benefit future work that builds on our findings. In our revised version, we now elaborate on two concrete directions in the Discussion section (changes in the PDF are shown in green):
>
> 1. Heuristic routing: where query relaxation counts are used when their signal is strong (for example, few ties remaining after a relaxation step), otherwise falling back to a neural model when query relaxation produces large tie groups. In this sense, query relaxation provides a notion of confidence based on how discriminative counts are for a given query. The challenge of implementing such an approach is determining precise strategies for choosing one approach over the other.
> 2. Rank-level ensembling, where normalized ranks or scores from query relaxation and neural models are combined using learned weights. This approach requires devising principled combination methods, as RELAX computes counts, which have a different interpretation than the normalized scores computed by neural models.
>
> **RC2:** We thank you for this important suggestion, which aligns with comments from rev. hT4K. To address this, we have conducted an additional experiment on path queries of length 2, where we observe performance at each relaxation step, which we now include in Appendix C. We observe that average MRR does not decrease as relaxation proceeds. Each step only refines the scores of entities with identical counts (tie groups, as defined in Section 4). In contrast, neural methods adjust scores for all entities at each traversal step, often via successive multiplication which can lead to error accumulation, particularly for longer path queries. This analysis shows that by adjusting the scores of entities in tie groups only, RELAX could reduce the chances of error accumulation that occur in neural models.
>
> **RC3:** We have added brief reminders of the query type notation in the experiments section, explicitly clarifying how 3p queries refer to path queries of length three, with references to Figure 1.
>
> **RC4:** We have standardized the terminology by using “query relaxation” for the general approach and RELAX only for our specific implementation, removing the term “relaxation-based reasoning” throughout the paper.

---

### Review · Reviewer_GR5S · 2026-01-19

**Summary Of Contributions:**

This paper shows that a simple heuristic baseline is able to solve some parts of the Complex Query Answering problem that existing neural methods cannot solve, and that there is potential for combining the approaches.

(CQA) In Complex Query Answering (CQA) one is given a knowledge graph (KG) and a question like the example in this paper: “What is the location of the baseball team for which Aaron Judge plays?”. These questions correspond to a small sub-graph of the KG where the answer could be found by traversing the graph from known anchor entities to unknown target entities, except that missing relations may preclude the existence of such a path. Neural approaches to CQA are state of the art according to existing benchmarks, but recent work (Gregucci et al. '25) showed that these benchmarks focus mostly on simple queries and proposed a new benchmark that is used here to deconstruct performance according to query complexity.

(RQ1) This paper proposes a simple query relaxation heuristic for CQA which is called RELAX. It ranks potential target/answer entities according to how many paths exist in the known graph that are similar to the query (except with a minimal number of entities or relations changed). In the first (RQ1) experiment the paper establishes that this heuristic and non-learned baseline can outperform some neural methods on some more complex query types, though the neural methods most often perform best.

(RQ2) Approaches produce a ranked list of answers and are evaluated via MRR (Mean Reciprocal Rank). The paper next compares approaches by asking how similar the ranked lists of answers are between RELAX and 3 neural approaches. It shows that there is significant dissimilarity between these lists: most answers are not shared between a pair of approaches, fewer answers are shared for (roughly speaking) higher complexity queries, and there is variation across datasets. The conclusion is that the errors of neural methods are complementary to the errors of RELAX.

(RQ3) Finally, the paper considers an oracle approach that combines RELAX with neural approaches to investigate how complementary they could be. It does this by picking the ranked answer list from either RELAX or a neural approach like QTO on a per-query basis. The results show consistent improvements in MRR across all query levels and datasets, so the paper concludes there is something to be gained here.

**Audience:**

Yes

**Audience Explanation:**

I think the audience for this paper includes people interested in CQA, but also those interested in interpretability of neural nets.

**Broader Impact Concerns:**

No concerns

**Claims And Evidence:**

Yes

**Claims Explanation:**

The claims in the paper are mostly well supported by experiments, but the paper does over-estimate the performance of the proposed RELAX baseline by claiming that it is closer in performance to the neural methods than it actually is (see 2nd part of RQ1 below).

> RQ1 (Comparable Performance): "Neural methods do not consistently perform better than query relaxation strategies. Across datasets and query structures, the two perform similarly, despite the latter involving no learning or optimization."

The first sentence is well substantiated (Tab 1), but the 2nd is not. I do not think it is fair to say that RELAX performs similarly to the neural approaches for the more complex queries (2i, 3i, pi, ip). In Table 1 it looks like it almost always underperforms them.

> RQ2 (Low answer overlap): "The top ranked answers from neural and relaxation-based approaches diverge substantially, indicating that they capture different patterns in the graph."

This claim is well substantiated by the IOU experiments (Fig 2).

> RQ3 (Complementarity): "Combining their answer sets consistently improves performance, suggesting that both contribute distinct and complementary reasoning signals."

This claim is well substantiated by Table 2.

**Requested Changes:**

* Do not claim that the RELAX approach matches the performance matches that of neural approaches across all the different configurations studied and more clearly emphasize where the RELAX approach struggles. That change does nothing to contradict the main point of the paper; I think the paper still clearly details a complementarity between RELAX and the studied neural approaches.

* The paper reports MRR improvements in relative terms instead of absolute terms. I think an improvement of +0.02 MRR from 0.6 is probably more difficult than +0.02 from 0.2 and relative percent improvement suggests the opposite. It may be better to present absolute improvement instead of or in addition to relative improvement.

---

> ### Author Response · Authors · 2026-02-06
>
> We thank the reviewer for their careful reading of the paper, their constructive feedback, and their recognition of the paper’s relevance to the CQA and interpretability communities. We have incorporated the requested changes (RC) in our updated version (changes are shown in green in the PDF), which we detail next:
>
> **RC 1:** We thank the reviewer for pointing this out and agree that the original wording could be misinterpreted. Our results do not imply that query relaxation matches neural methods on complex queries; rather, they show that neural methods do not exhibit clear or consistent superiority across all datasets and query structures. We have therefore rephrased the corresponding statement in the introduction (now “Unclear superiority”) to emphasize the absence of a uniformly dominant approach, in line with the analysis in our Experiments section.
>
> **RC 2:** To provide a more informative view of performance gains, we have updated Tables 2, 6, and 7 to include absolute MRR improvements, in addition to the current relative ones. This allows readers to better assess the magnitude of the gains across different performance levels.

---

> > ### Comment · Reviewer_GR5S · 2026-02-07
> >
> > The revised version of the paper clarifies the first claim. Now I think the claims are all well supported and I adjusted my review to reflect that.

---

### Review · Reviewer_hT4K · 2026-01-27

**Summary Of Contributions:**

This paper systematically analyzes the strength of query relaxation in answering complex knowledge graph-based queries. The thorough empirical results demonstrate that the SOTA neural approaches do not outperform this simple strategy consistently, and there is no clear overlap in the rankings proposed by the neural methods and this training-free strategy. Consequently, the authors show that combining the two directions help to further boost the performance of complex query answering by a clear margin.

**Audience:**

Yes

**Audience Explanation:**

The empirical results of this paper show that the simple, training-free strategy of query relaxation is a strong baseline in complex query answering. This is clearly an important finding for the community, where future work, especially those neural methods, should sincerely consider this baseline in their reports.

**Claims And Evidence:**

Yes

**Claims Explanation:**

- The proposed strategy of query relaxation, albeit not entirely novel, is easy to follow and clearly demonstrated by Example 2.
- The three research questions studied in Section 5 are strongly motivated and built upon sequentially.
- The empirical results to the three research questions are clearly presented and accurately supported by the figures and tables in Section 5.

**Requested Changes:**

It would be great to include some error analysis of the query relaxation strategy. Since the relaxation happens step by step, as explained in Section 4, it is important to check whether this would lead to error escalation as more relaxation is allowed.

---

> ### Author Response · Authors · 2026-02-06
>
> We thank the reviewer for their time, their positive assessment of our work, and their constructive feedback. We have updated our submission in response to the requested changes. We agree that error analysis is important in our work, which is reflected by our experiments where we analyze specific query types where query relaxation brings larger or smaller improvements (Table 2). We have complemented these results with the requested change, which brings additional insights on the effect of individual relaxation steps.
>
> During query relaxation, we expect performance at each relaxation step to result in similar performance (instead of error always increasing), because each step modifies only the scores of entities that have received identical counts (referred to as tie groups in Section 4). To confirm this hypothesis, we carry out experiments with path queries of length two (denoted as 2p) and measure MRR at each relaxation step, and obtain the following results:
>
> | Relaxation step | FB15k237+H | NELL995+H | ICEWS18+H |
> | --------------- | ---------- | --------- | --------- |
> | 1               | 0.0315     | 0.0428    | 0.0148    |
> | 2               | 0.0315     | 0.0428    | 0.0148    |
> | 3               | 0.0401     | 0.0565    | 0.0160    |
>
> These results show that for 2p queries, the second step maintains MRR, indicating that breaking ties at this step has minimal effect for 2p queries, while in all cases, the last relaxation step results in further increasing MRR, supporting the stability of the relaxation strategy. We have added this analysis to Appendix C (changes shown in green in the PDF).

---

### Decision · Action_Editor_WDTH · 2026-02-28

**Recommendation:** Accept as is

**Audience:**

Yes

**Audience Explanation:**

Readers interested in Complex Query Answering, interpretability of neural networks, and hybrid reasoning would find this work interesting.

**Claims And Evidence:**

Yes

**Claims Explanation:**

After a minor clarification, all reviewers agree that the claims are supported by clear evidence.